# A Preliminary Study on Microbiota Characteristics of Bronchoalveolar Lavage Fluid in Patients with Pulmonary Nodules Based on Metagenomic Next-Generation Sequencing

**DOI:** 10.3390/biomedicines11020631

**Published:** 2023-02-20

**Authors:** Qian Yuan, Xiaojin Wang, Zhanglin Li, Wenzhuo Guo, Hua Cheng, Qingdong Cao

**Affiliations:** Department of Thoracic Surgery, The Fifth Affiliated Hospital of Sun Yat-Sen University, Zhuhai 519000, China

**Keywords:** mNGS, pulmonary nodules, microbiota, BALF

## Abstract

Background: The characteristics and roles of microbes in the occurrence and development of pulmonary nodules are still unclear. Methods: We retrospectively analyzed the microbial mNGS results of BALF from 229 patients with pulmonary nodules before surgery, and performed a comparative analysis of lung flora between lung cancer and benign nodules according to postoperative pathology. The analysis also focused on investigating the characteristics of lung microbiota in lung adenocarcinomas with varying histopathology. Results: There were differences in lung microbiota between lung cancer and benign lung nodules. Bacterial diversity was lower in lung cancer than in benign lung nodules. Four species (*Porphyromonas somerae*, *Corynebacterium accolens*, *Burkholderia cenocepacia* and *Streptococcus mitis*) were enriched in lung cancer compared with the benign lung nodules. The areas under the ROC curves of these four species were all greater than 0.6, and the AUC of *Streptococcus mitis* was 0.702, which had the highest diagnostic value for differentiating lung cancer from benign lung diseases. The significantly enriched microbiota varied with the different pathological subtypes of lung adenocarcinoma. *Streptococcus mitis*, *Burkholderia oklahomensis* and *Burkholderia latens* displayed a trend of increasing from the benign lung disease group to the AIS group, MIA group and IAC group, whereas *Lactobacillus plantarum* showed a downward trend. Conclusion: Changes in the abundance of lung microbiota are closely related to the development of infiltrating adenocarcinoma. Our findings provide new insights into the relationship between the changes in lung microbiota and the development of lung cancer.

## 1. Introduction

Lung cancer is one of the cancers with the highest morbidity and mortality worldwide. The latest data show that lung cancer has the highest incidence and mortality in China, which poses a great threat to human health [1,2]. Although low-dose spiral CT has been widely used, most lung cancer patients are still diagnosed with advanced lung cancer at the first visit, missing the opportunity for early radical surgery [3]. Adenocarcinoma in situ (AIS), minimally invasive adenocarcinoma (MIA) and invasive adenocarcinoma (IAC) are different pathological subtypes of lung adenocarcinoma [4]. Although lung adenocarcinoma mainly appears as pulmonary nodules on imaging, the prognosis varies greatly. The 5-year survival rate of AIS and MIA is 100%, while that of invasive adenocarcinoma is 67% [5]. Besides the pathological findings, there is no valid biomarker to distinguish between lung adenocarcinomas with different pathological subtypes.

Numerous studies have shown a direct causal relationship between cancer and microbes [6]. For example, H. pylori and Fusobacterium promote the development of gastric cancer and colon cancer, respectively [7,8]. It has been reported that lung cancer is also correlated with lung microbes, but the distribution characteristics and roles of lung microbes in the occurrence and development of pulmonary nodules are still unclear. Current research on the relationship between microbes and lung cancer mainly focuses on microbes in sputum, saliva, feces and lung tissue [9]. Sputum, saliva, feces and other samples are easily contaminated by the external environment of the host, while lung tissue microorganisms are not easy to obtain and are difficult to be used as a means of preclinical diagnosis and treatment [10]. BALF can accurately reflect the microenvironment of lung cancer, and its contamination risk of the upper respiratory tract, sampling accuracy and invasiveness are acceptable [11]. At present, there are few studies on the relationship between microbes in BALF and lung cancer. The potential role of the microbiota in lung cancer susceptibility remains to be determined. Several studies have confirmed the existence of some unique flora in BALF [12,13]. Most previous studies have compared the microbial composition of BALF between typical lung cancer patients and healthy individuals or benign diseases. However, the differences in pathological types, stages, tumor sites and the solid component of the ground glass nodule may affect microbial characteristics [14]. The sample size of current studies is small, and there is no study on the microbial characteristics of lung adenocarcinoma at different growth stages (the process from AIS to IAC). Existing studies are mostly based on 16S rRNA sequencing, which can only distinguish different types of bacteria at the genus level, and the results are difficult to generalize clinically. mNGS can identify microorganisms at the species level and accurately analyze the metabolic pathways of marker species at the gene level [15].

This study retrospectively analyzed the characteristics of and differences in pulmonary microbiota in patients with pulmonary nodules at the species level based on metagenomic sequencing. Additionally, we sought to correlate the changes in the lung microbiota with the development of invasive cancer. Our results explore the specific microbial markers for the prediction of lung cancer and provide new ideas for the diagnosis and treatment of lung cancer.

## 2. Material and Methods

### 2.1. Study Population

Patients who presented with suspicious nodules on CT images and underwent metagenomic next-generation sequencing of microorganisms in bronchoalveolar lavage fluid before surgery at the Fifth Affiliated Hospital of Sun Yat-sen University (Zhuhai, China) between July 2020 and June 2022 were enrolled in this retrospective study. Pathology obtained by video-assisted thoracoscopic surgery (VATS) was the most important criterion for patient inclusion in the study because it confirmed the diagnosis and validated the histopathological classification. Demographic and clinical data were obtained from each participant, including age, sex, smoking status, pulmonary function tests, family history and pathological types.

Inclusion criteria: (1) age > 18 years old; (2) the pathological diagnosis was lung cancer or benign lung disease confirmed by surgery; (3) microbiological analysis of BALF was performed via mNGS before surgery and we could obtain complete data on the relative abundance of microbes for each patient.

Exclusion criteria: (1) participants with lung cancer who had undergone a second operation or manifested other basic pulmonary diseases; (2) suffering from immune-compromising diseases, such as human immunodeficiency virus (HIV) or other cancers in addition to lung cancer; (3) participants received glucocorticoid or antibiotic treatment 3 months or less before sample collection; (4) patients had received chemotherapy, radiation therapy or other treatments for lung cancer before surgery.

### 2.2. Sample Collection and DNA Extraction

Bronchoscopy was performed by an experienced clinician on all eligible participants before treatment. The bronchoscopy was performed according to the standardized protocol to minimize oral contamination [16]. The bronchoscope was quickly wedged into the segmental bronchus where the pulmonary nodule was located. Then, 2 mL BALF was collected from each patient and placed on ice before storing it at −80 °C within 30 min. Then, 2 mL microcentrifuge tubes with 0.6 mL of sample and 0.5 mL glass beads were attached to a horizontal platform on a vortex mixer and agitated vigorously at about 3200 rpm for 20 min. After agitation, 0.3 mL of the sample was separated into a new 2 mL microcentrifuge tube, and DNA was extracted using a TIANamp Micro DNA Kit (DP316, Tiangen Biotech, Beijing, China) according to the manufacturer’s instructions.

### 2.3. Library Preparation and Sequencing

DNA libraries were established by breaking, end repair, adapter ligation and PCR amplification of the extracted DNA. The quality of the DNA libraries was assessed using an Agilent 2100 Bioanalyzer (Agilent Technologies, Santa Clara, CA, USA) combined with quantitative PCR to measure the adapters before sequencing. Qualified DNA libraries were sequenced on the BGISEQ-50 platform (BGI, Shenzhen, China). After sequencing data were offloaded, low-quality data and reads with a length less than 35 bp were removed to obtain high-quality data, followed by computational subtraction of human host sequences mapped to the human reference genome (hg19) using Burrows–Wheeler alignment. The data remaining after the removal of low-complexity reads were classified by simultaneous alignment into four microbial genome databases consisting of viruses, bacteria, fungi and parasites. The classification reference databases were downloaded from NCBI (ftp://ftp.ncbi.nlm.nih.gov/genomes/ 10 August 2022), containing 10,989 bacteria, 1800 viruses, 1179 fungi and 282 parasites related to human diseases [17].

### 2.4. Statistical Analysis

We converted read abundance to percentages based on the total number of high-quality mapped sequences for each sample at the species and genus levels. These normalized data were used for all subsequent statistical analyses. All statistical analyses were performed using R (version 4.2.1, R Foundation for Statistical Computing, Vienna, Austria) software and SPSS (version 25.0, IBM Corporation, Armonk). Qualitative data were compared between groups using the Wilcoxon rank-sum test or an independent t-test, and quantitative data were determined via crosstabs with the chi-square test. The Wilcoxon rank-sum test was used to compare alpha diversity measures. NMDS (non-metric multidimensional scaling) was used to compare beta diversity measures. ANOSIM was performed to test for the statistical significance of beta diversity. LEfSe analysis was used to estimate microbiota with differential abundance among the groups. For the differential genera obtained through LEfSe analysis, we used the receiver operating characteristic curve (ROC) to estimate the diagnostic value. A two-sided *p* value of less than 0.05 was considered statistically significant.

## 3. Results

### 3.1. Baseline Characteristics of Study Subjects

Our retrospective study included 229 patients, including 192 patients with lung cancer and 37 patients with benign pulmonary nodules. The median age was 55.24 ± 12.94 in the lung cancer group and 49.32 ± 12.30 in the control group. The baseline data of the two groups were similar except for age. However, the difference in age is consistent with the epidemiology of lung cancer, and the confounding factors can be basically excluded. The patients in the lung cancer group had mainly non-small cell lung cancer, including 185 cases of adenocarcinoma and 6 cases of squamous cell carcinoma. The subjects included in our study had mainly early-stage lung cancer. Adenocarcinoma accounted for 95% of the enrolled patients, which included 31 patients with adenocarcinoma in situ, 51 patients with minimally invasive adenocarcinoma and 101 patients with invasive adenocarcinoma. The clinical characteristics of all subjects are shown in Table 1.

### 3.2. Biodiversity between Lung Cancer and Benign Lesions

α-Diversity reflects the richness and evenness of microbial communities by calculating the Shannon and Simpson indexes. The Shannon index is positively correlated with the richness and evenness, while the Simpson index is negatively correlated with them. The Wilcoxon rank-sum test was performed to determine the significance of the differences in the index values between the two groups. The Simpson index of the lung cancer group was significantly higher than that of the control group (*p* = 0.0097), while the Shannon index showed no significant difference between the two groups (*p* = 0.14). Our results demonstrated that the richness and evenness of lung microbiota in lung cancer patients were lower than in benign lesions (Figure 1A,B).

β-Diversity is a measure of the difference in the overall microbiota composition between groups. A non-metric multidimensional scaling (NMDS) plot was used to visualize the overall microbiome dissimilarity measured by the Bray–Curtis distance. The more similar samples are in species abundance and composition, the closer they are in the NMDS plot. The NMDS plot demonstrated that no obvious separation between lung cancer and benign disease was observed, suggesting that the composition and structure of microbiota were similar (Figure 1C). ANOSIM analysis also revealed that there were no significant differences in β-diversity between the two groups (*p* > 0.05) (Figure 1D). However, when the lung cancer group was further divided into the squamous cell carcinoma group and the adenocarcinoma group, the results of the NMDS and ANOSIM analysis revealed that no significant differences in β-diversity among the three groups were observed, which proved that different pathological types could affect the composition of lung microbiota (Figure 1E,F).

The top 10 species with the largest relative abundance in each sample were selected as the dominant flora based on the species-level relative abundance table. Figure 2A shows that the composition of the top 10 species in relative abundance was similar between lung cancer and benign lesions. Among these 10 species, the relative abundance of *Streptococcus mitis* (Wilcoxon test, *p* = 0.0001) (Figure 2B) and *Porphyromonas somerae* (Wilcoxon test, *p* = 0.029) (Figure 2C) in lung cancer was significantly higher than that in benign lung diseases.

### 3.3. Analysis of Differences in the Microbiota Community between Lung Cancer and Benign Lesions

The results of the diversity analysis showed that there were some differences in lung microbiota between lung cancer and benign pulmonary nodules. LEfSe analysis was used to further compare the microbiota, with significantly different abundances among different groups to identify potential microbial biomarkers. The results showed that 23 bacteria with significant abundance differences were identified at the species level (Figure 3A). *Streptococcus mitis*, *Porphyromonas somerae*, *Corynebacterium accolens* and *Burkholderia cenocepacia* were enriched in lung cancer. *Mycobacterium kansasii*, *Klebsiella pneumonia* and *Roseomonas mucosa* were enriched in benign lesions. *Streptococcus mitis* and *Porphyromonas somerae* were consistent with the differential bacteria identified in our analysis of the top 10 dominant bacteria in abundance, which also proved the stability of our statistical results. We also compared the microbiota compositions among different pathological types of lung cancer, including between the squamous cell carcinoma group and benign pulmonary nodule group (Figure 3B), the adenocarcinoma group and squamous cell carcinoma group (Figure 3C), and the adenocarcinoma group and benign pulmonary nodule group (Figure 3D). We found that different pathological types of lung cancer were significantly enriched with different microbiota. *Streptococcus mitis*, *Porphyromonas somerae*, *Corynebacterium accolens* and *Fusobacterium nucleatum* were enriched in adenocarcinoma. *Geobacillus thermodenitrificans* was enriched in squamous cell carcinoma.

There is no significant demographic difference between lung cancer and benign lung lesions, but smoking has been confirmed as an important risk factor for lung cancer. We further analyzed the differential microbiota between smoking and non-smoking patients with lung cancer. Taxa with significant abundance differences between smokers and non-smokers were identified. *Mycobacterium kansasii*, *Streptococcus parasanguinis* and *Atopobium parvulum* were enriched in smoking patients. *Cutibacterium acnes*, *Cutibacterium acnes* and *Cutibacterium acnes* were enriched in non-smoking patients (Figure 3E).

### 3.4. Potential Microbe Biomarkers for Lung Cancer

Based on the bacterial biomarkers screened in the LEfSe analysis, we determined which potential biomarkers had the best diagnostic value by drawing the ROC curve and calculating the AUC value. We found that the AUC values of *Streptococcus mitis*, *Porphyromonas somerae*, *Corynebacterium accolens* and *Burkholderia cenocepacia* were all greater than 0.6, and the AUC value of *Streptococcus mitis* was 0.702, which had the highest diagnostic value and certain accuracy in differentiating patients with lung cancer from those with benign lung diseases (Figure 3F). Our finding is consistent with previous studies. However, previous studies only reported that *Streptococcus* may be a potential biomarker for lung cancer. They did not specify the *Streptococcus* at the species level. In our study, a specific species of *Streptococcus* was found through metagenome sequencing.

### 3.5. Differences in Microbiota According to Different Histopathological Lung Adenocarcinomas

According to the WHO histological classification of lung tumors in 2021, adenocarcinoma patients were further divided into three groups: 31 patients in the adenocarcinoma in situ group (AIS group), 51 patients in the minimally invasive adenocarcinoma group (MIA group) and 101 patients in the invasive adenocarcinoma group (IA group). This article attempts to describe the characteristics of and changes in lung microbiota from the perspective of the development of adenocarcinoma.

#### 3.5.1. Biodiversity Analysis

We first calculated the Shannon index and Simpson index to estimate the α-diversity of the microbial communities among the four groups. The results showed that no significant difference in the Shannon index among the four groups was identified, and no significant difference in the Simpson index among the different lung cancer groups was identified. However, the Simpson index of the benign lung disease group was significantly lower than that of the IAC group. Our results demonstrated that no significant difference in α-diversity among the three different pathological subgroups of lung adenocarcinoma was observed. The alpha diversity of the IAC patients was lower than that of patients with benign pulmonary nodules (Figure 4A,B). β-Diversity was based on NMDS analysis and ANOSIM analysis, and the results indicated that no significant difference in the overall composition of the microbiota among the four groups was observed (*p* > 0.05) (Figure 4C,D).

#### 3.5.2. Significant Differential Microbiota Compositions

LEfSe analysis revealed that the dominant lung microbiota was specific to the histopathological subtypes of lung adenocarcinoma. There were six, five, four and three microorganisms at the species level that were significantly different in the HP, AIS, MIA and IAC groups, respectively (Figure 5A). *Serratia grimesii*, *Pseudochrobactrum saccharolyticum*, *Acinetobacter schindleri*, *Proteus hauseri*, *Methylobacillus flagellatus* and *Alcaligenes aquatilis* were enriched in the AIS group. *Serratia grimesii*, *Pseudochrobactrum saccharolyticum*, *Acinetobacter schindleri*, *Proteus hauseri*, *Methylobacillus flagellatus* and *Alcaligenes aquatilis* were enriched in the MIA group. *Streptococcus mitis*, *Burkholderia oklahomensis*, *Burkholderia latens* and *Enterococcus faecalis* were enriched in the IAC group. We further analyzed the abundance of eighteen different species across the four groups using the Wilcoxon rank-sum test. The relative abundance bar chart is used to show the differences more visually.

Among the eighteen discriminative species, *Streptococcus mitis* (Figure 5B), *Burkholderia oklahomensis* (Figure 5C) and *Burkholderia latens* (Figure 5D) showed a trend of increasing from the benign lung disease group to the AIS group, MIA group and IAC group, whereas *Lactobacillus plantarum* decreased (Figure 5E). Our results suggest that changes in four differential taxa may have some connection with the development of lung adenocarcinoma, which requires further experimental studies.

## 4. Discussion

For a patient with pulmonary nodules, it is very important to be able to determine accurately whether the nodule is benign or malignant before treatment. If it is a malignant nodule, it is also extremely important to judge accurately the degree of invasion before surgery for the selection of surgical methods. These are the strategies clinicians need to consider in the treatment of pulmonary nodules. Therefore, we tried to solve this problem from the perspective of microbial markers.

The characteristics and roles of microbes in the occurrence and development of pulmonary nodules have not been elucidated, and different research results may be obtained due to the different environments, microbial sampling sites, sampling tissues, etc. The sample size of the existing studies is small, and the research samples are all typical lung cancer cases, so the results are limited and difficult to use in clinical practice. Most studies use 16S rRNA gene sequencing, which can only analyze the microbial composition at the genus level [18]. Our study aimed to investigate the microbial characteristics of and differences in microbiota in BALF from patients with pulmonary nodules based on mNGS at the species level. To the best of our knowledge, there is no study on the relationship between the lung microbiota and the growth of lung adenocarcinomas with varying histopathology. We have the largest sample size of lung adenocarcinoma based on mNGS to date, which covers all pathological processes of lung adenocarcinoma. Our study is the first to compare the composition of and differences in the pulmonary microbiota among different histopathological types of lung adenocarcinoma.

We first analyzed the overall microbiota diversity between lung cancer and benign pulmonary nodules. We found that the α-diversity of patients with lung cancer was lower than that of patients with benign nodules, but there was no significant difference in β-diversity between the two groups. The results of previous studies were heterogeneous due to the different types of samples collected, so we compared the studies that directly analyzed the microbiota of lung cancer only using BALF. Zeng et al. [19] showed significantly higher α-diversity in lung cancer than in benign nodules, but no significant difference in β-diversity was observed. However, their study did not exclude patients with underlying lung diseases such as pneumonia, COPD and pulmonary fibrosis, which may have had a certain impact on the results [20]. Zhuo et al. [13] and Lee et al. [12] found that there was no difference in α- and β-diversity between lung cancer patients and the control group, but Zhuo et al. compared cancerous lungs with contralateral non-cancerous lungs. Wang et al. [21] concluded that the microbiota diversity decreased in lung cancer patients, which is similar to our results, but the samples in the control group were from healthy individuals. The differences in results may be due to differences related to the living environment, the number of samples collected, the control group and the analysis of sequencing data. Our study subjects were patients with pulmonary nodules that were difficult to distinguish as benign or malignant on imaging, which may be one of the reasons why our results showed no significant difference in the overall microbiota composition (β-diversity) between patients with lung cancer and those with benign pulmonary nodules.

We performed a differential analysis of the top 10 species in relative abundance. We found that the relative abundance of *Streptococcus mitis* and *Porphyromonas somerae* in lung cancer was significantly higher than that in benign lung diseases. The same results were obtained in the LfFSe analysis, which proved the stability and reliability of our statistical analysis. ROC analysis showed that *Streptococcus mitis*, *Porphyromonas somerae*, *Corynebacterium accolens* and *Burkholderia cenocepacia* may be potential biomarkers for lung cancer, and *Streptococcus mitis* has the greatest value in differentiating between benign and malignant pulmonary nodules. *Streptococcus* has been identified as a major marker associated with lung cancer in several previous studies, which has been reported in saliva [22], sputum [23,24], BALF [19] and lung tissue [25]. Although there are some differences in these studies, they suggest that *Streptococcus* plays a non-negligible role in the microbial environment of lung cancer. All the above studies used bacterial 16S rRNA PCR amplification. Our study is based on mNGS, which can identify unknown microorganisms and classify the microbiome at the species level. We found that *Streptococcus mitis* may be an actual species or strain involved in carcinogenesis. According to basic experimental reports, changes in the IL-23/IL-17 axis [26] are well known in the pathogenesis of autoimmune diseases and tumors. *Streptococcus mitis* can induce the transcription of IL-1β, IL-6, IL-10 and IL-23, activation of Th17 and expression of immune checkpoint PD-L1, thereby promoting the development and invasion of tumors. Whether the enrichment of a large number of *Streptococcus* in lung cancer is the cause or the result of tumors is an unresolved question. Our results indicated that *Streptococcus mitis* might provide a new and more accurate research target, and we need to further explore its effect on the development and immunotherapy of lung cancer. The microbial profiles of squamous cell carcinoma and adenocarcinoma were analyzed, and the results indicated no significant differences in β-diversity between patients with different pathological types of lung cancer and benign lung diseases, which is consistent with the results of previous studies [27]. However, we found different bacterial flora in different pathological types of lung cancer at the species level. We found that *Streptococcus mitis* and *Geobacillus thermodenitrificans* may be microbial markers for adenocarcinoma and squamous cell carcinoma, respectively. This has not been reported in previous studies. Due to the small number of patients with squamous cell carcinoma in our study, further studies with more squamous cell carcinoma patients are needed. In addition, our study showed that smoking patients had significantly different microbiota compared with non-smoking patients. This may imply that environmental factors such as smoking can increase the risk of lung cancer by altering the microbial composition.

Adenocarcinoma is the most common pathological type of NSCLC. There are three main pathological types, including adenocarcinoma in situ, minimally invasive adenocarcinoma and invasive adenocarcinoma. Although the above lesions are mainly ground glass nodules on CT images, the surgical methods and prognosis are quite different [28]. Therefore, it is very important to determine the degree of malignancy of pulmonary nodules before an operation. At present, no specific biological markers for invasive adenocarcinoma have been found. Our samples covered the entire development process of lung nodules from carcinoma in situ to invasive adenocarcinoma. To identify the biomarkers in the development of lung adenocarcinoma, we attempted to explore the relationship between lung microbiota and the development of invasive cancer using mNGS. There were no significant differences in α-diversity and β-diversity among different histopathological types of adenocarcinomas, except that the α-diversity of the IAC group was lower than that of benign pulmonary disease. However, different pathological subtypes of lung adenocarcinoma have their own enriched microbiota. Our results showed that *Streptococcus mitis* was mainly enriched in invasive adenocarcinoma, which again proved the stability of our statistical analysis results. *Streptococcus mitis*, *Burkholderia oklahomensis* and *Burkholderia latens* displayed a trend of increasing from the benign lung disease group to the AIS group, MIA group and IAC group, whereas *Lactobacillus plantarum* decreased. The results suggested that *Streptococcus mitis*, *Burkholderia oklahomensis* and *Burkholderia latens* might be risk factors for the development of invasive adenocarcinoma, while *Lactobacillus plantarum* might be a protective factor. Interestingly, a gradual shift in the microbiota distribution from gastritis to preneoplastic lesions to cancer was also reported in a previous study by Aviles-Jimenez et al. [29]. A study by Flemer B et al. [30] compared the microbiota between colorectal cancer patients, polyp patients and healthy people. The microbiota showed significant differences between colorectal cancer patients and healthy people, and between mucosal biopsies from patients with polyps and healthy people, suggesting that specific microbiota of colorectal cancer are already present and may be involved in the early stages of cancer development. Liu’s [31] study first reported a gradual “shift” in *Streptococcus* from “healthy” to non-cancerous to cancerous samples. It is worth mentioning that we found, for the first time, a gradual “shift” in the lung microbiota profile from the benign lung disease group to the AIS group, MIA group and IAC group. The DNA double-strand break mechanism caused by certain bacteria isolated from cancerous sites can lead to carcinogenesis. *Streptococcus* may be invasive, inducing cytokine and inflammatory responses that promote cancer development [32]. We guess that microbial changes may take precedence over local tumor changes; in particular, the dynamic changes in *Streptococcus mitis* may be closely related to the occurrence and development of lung cancer, thus helping to determine the stage of the disease. At the same time, more large-scale dynamic observations are needed in the future. Currently, studies on metabolomics and biomarkers for the early diagnosis of lung cancer mainly focus on expiratory metabolomics, blood metabolomics, urine metabolomics, fecal metabolomics and tissue metabolomics. However, the above metabolomics are still in basic research, and it is still necessary to explore biomarkers for the early diagnosis of lung cancer with high sensitivity and specificity [33]. There are few studies on the metabolomics of alveolar lavage fluid at home and abroad. Therefore, we will further study the relationship between bacterial metabolites in alveolar lavage fluid and the occurrence and development of lung cancer based on our findings thus far, in order to find screening markers for lung cancer that can be extended to clinical application.

Our study has some limitations: (1) samples of healthy people were not collected as a control group, and the results may lack representation. (2) Since the patients with pulmonary nodules included in the retrospective analysis were patients with a high possibility of malignancy, fewer patients with benign pulmonary diseases were collected. (3) This study is a cross-sectional study. At present, we have only found potential microbial markers, and there is still a lack of mechanism studies in cell and animal experiments to further confirm whether the discovered microbial differentials can be used as biomarkers. Further population-based cohort studies are also required for clinical application.

## 5. Conclusions

Based on postoperative pathology, we conducted a detailed study on the diversity of and differences in lung microbiota in patients with pulmonary nodules. There were differences in pulmonary microbiota between patients with lung cancer and those with benign pulmonary nodules. The diversity of microbiota in patients with lung cancer was lower than that in patients with benign pulmonary nodules. There were significant differences in microbiota among lung cancer patients with different pathological types and smoking statuses. The changes in the abundance of lung microbiota are related to the development of lung adenocarcinoma with different histopathological features. Our study offers a complete picture and analysis of the lung flora in patients with pulmonary nodules, and our findings provide new targets for the diagnosis and treatment of lung cancer, pending our further basic research.

## Figures and Tables

**Figure 1 biomedicines-11-00631-f001:**
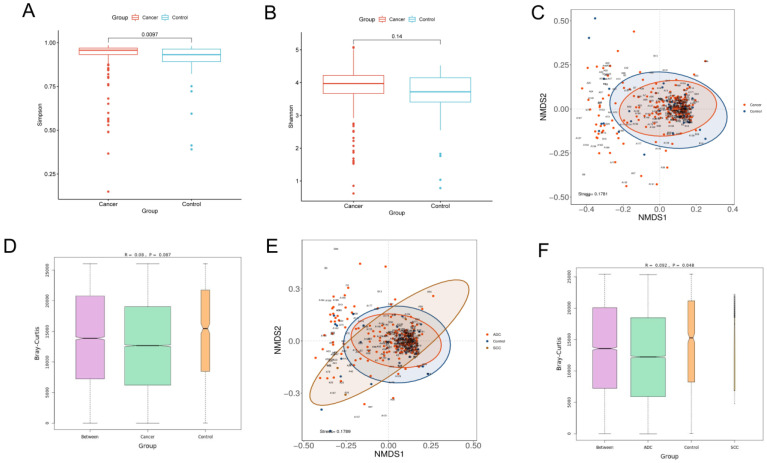
Diversity analysis for microbiota in BALF. Alpha diversity analysis between lung cancer and benign pulmonary nodules based on the (**A**) Simpson index (Wilcoxon rank−sum test, *p* < 0.05) and (**B**) Shannon index (Wilcoxon rank−sum test, *p* > 0.05); beta diversity analysis via NMDS based on the Bray−Curtis distance (**C**). ANOSIM analysis was used as a statistical test for beta diversity (**D**). The R value obtained in ANOSIM analysis was between −1 and 1. A value closer to 1 indicates that the difference between groups is greater than that within groups. When the *p* value is less than 0.05, the reliability of the test is high. Longitudinal seating represents the Bray−Curtis distance. Comparison of beta diversity analysis among adenocarcinoma, squamous carcinoma and benign lesions visualized in the NMDS plot (**E**). ANOSIM analysis was used as a statistical test for beta diversity. ANOSIM analysis box plot (**F**).

**Figure 2 biomedicines-11-00631-f002:**
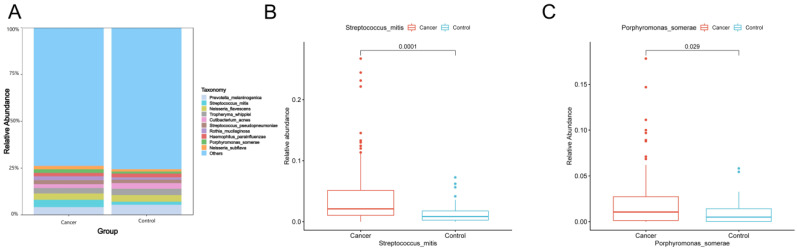
(**A**) Dominant species of lung cancer and control group. At the species level, the top 10 species with the largest relative abundance in the two groups are shown, with “Others” representing other species detected. The vertical axis shows the proportion of species in relative abundance, while the horizontal axis shows the sample grouping. The species category corresponding to each color block is shown in the legend on the right. (**B**,**C**) Box plots showing significant differences in abundance at the species level (Wilcoxon rank-sum test, *p* < 0.05). The horizontal axis indicates the sample grouping, whereas the vertical axis indicates the relative abundance of the corresponding species.

**Figure 3 biomedicines-11-00631-f003:**
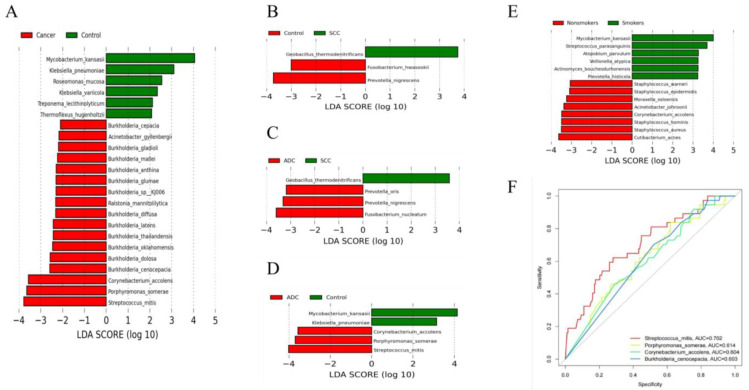
(**A**) Differentially abundant flora between lung cancer and benign pulmonary nodules identified by LEfSe; (**B**–**D**) differentially abundant flora among adenocarcinoma, squamous carcinoma and benign pulmonary nodules identified by LEfSe; (**E**) differentially abundant flora between smokers and non-smokers identified by LEfSe. Species with LDA values greater than the set point are shown. (**F**) Receiver operating characteristic (ROC) curves based on four significant differential species to distinguish lung cancer and benign pulmonary nodules.

**Figure 4 biomedicines-11-00631-f004:**
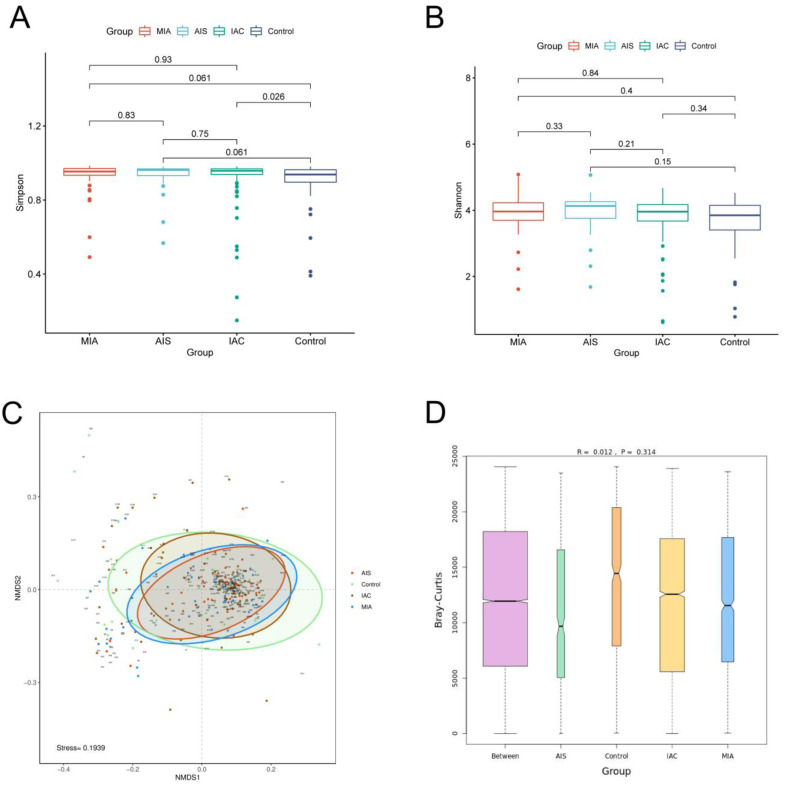
Diversity analysis for microbiota among four groups based on varying histopathology. Alpha diversity analysis among four groups based on the (**A**) Simpson index and (**B**) Shannon index; (**C**) beta diversity analysis via NMDS based on the Bray−Curtis distance. (**D**) ANOSIM analysis box plot.

**Figure 5 biomedicines-11-00631-f005:**
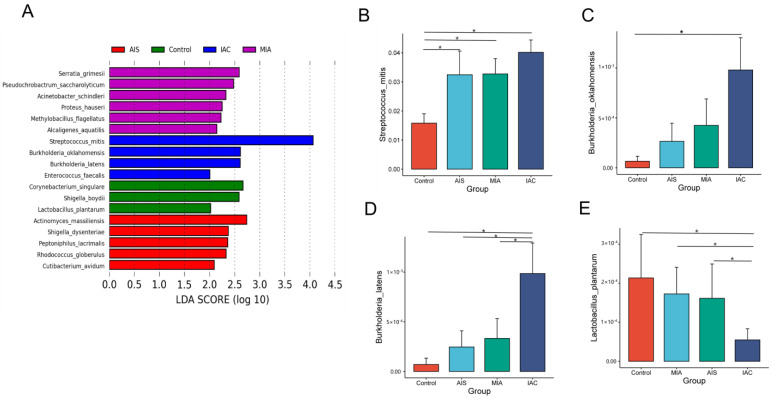
(**A**) Differentially abundant microbiota among four groups based on varying histopathology identified by LEFSE; (**B**–**E**) bar chart showing significant differences in abundance at the species level. The Wilcoxon rank-sum test was used as a statistical test, and “*” indicates a significant difference between two groups (*p* < 0.05).

**Table 1 biomedicines-11-00631-t001:** Baseline clinical characteristics of the study subjects.

Variables	Lung Cancer (*n* = 192)	Benign Lesions (*n* = 37)	*p*
Age	55.24 (12.94)	49.32 (12.30)	0.014
Sex			0.131
Male	78 (40.6)	20 (54.1)	
Female	114 (59.4)	17 (45.9)	
BMI	23.55 (3.63)	24.95 (12.09)	0.489
Pulmonary function test	(*n* = 80)	(*n* = 14)	
FVC (%)	98.70 (18.67)	105.02 (12.22)	0.226
FEV1 (%)	97.17 (8.6)	98.75 (11.52)	0.55
DLCO (%)	86.8 (14.14)	79.1 (22.65)	0.095
Smoking status			
Never	153 (79.7)	30 (81.1)	0.846
Ever	39 (20.3)	7 (18.9)	
Histology			
ADC	183 (95.3)	-	-
SCC	6 (3)	-	-
Others	3 (1.6)	-	-
Tumor stage			
0	31 (16.15)	-	-
I	135 (70.31)	-	-
II	6 (3.13)	-	-
III	11 (5.7)	-	-
IV	7 (3.6)	-	-

Continuous variables are presented as mean (standard deviation). Categorical variables are expressed as number (%). BMI, body mass index; FVC, forced vital capacity; FEV1, forced expiratory volume; DLCO, diffusing capacity of the lung for carbon monoxide; % pred, percentage of the predicted value; ADC, adenocarcinoma; SCC, squamous cell carcinoma.

## Data Availability

The datasets supporting the conclusions of this article are included within the article; further inquiries can be directed to the corresponding authors.

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
