# Peer review of "A Preliminary Study on Microbiota Characteristics of Bronchoalveolar Lavage Fluid in Patients with Pulmonary Nodules Based on Metagenomic Next-Generation Sequencing"

_biomedicines, 2023, doi:10.3390/biomedicines11020631_

Round 1
Reviewer 1 Report
Although the quality of study is good some language corrections are needed and moreover I have few remarks on the content.
1- references are no reported correctly.
2-Abstract is too long more of 250 wors
3-Conclusion are no supported by data. the data shown is not robust enough to conclude that Streptococcus mitis may be used as marker for lung cancer. This statement should be removed.
Reviewer 2 Report
This is a relevant paper concerning the Study on Microbiota Characteristics of Bronchoalveolar Lavage Fluid in Patients with Pulmonary Nodules Based on Metagenomic Next-generation Sequencing.
The authors conducted a detailed study on the diversity and differences of lung microbiota in patients with pulmonary nodules. They claimed that there are differences in pulmonary microbiota between patients with lung cancer and benign pulmonary nodules.
The study is relevant and the results support the conclusion.
There are some issues that should be solved before the work is published.
References are not correctly introduced throughout the entire document.
Pag 2 line 76: There is an extra space between “Sequencing of” and “microorganism”
Figure 1 Pag 5: Figure 1 is very difficult to read. It does not have enough quality.
Throughout the paper microorganisms’ names are not correctly written. They should be according to taxonomic rules. (Italic, first name with caps ad the rest not). It is not Streptococcus_mitis it is Streptococcus mitis (in italics)
Figure 4 caption is not on the same page as the Figure.
Pag 3 line 105: library should be Library
Pag 4 Table 1: There are differences in the format. Explain (in the caption or in the end) what are the numbers between parentheses and sometimes there are spaces and sometimes not.
Pag 4 line 173: Where it is written “control group(p=0.0097), while” it should be written “control group (p=0.0097).
Page 8/9: Why is Figure 4/5 in bolt in the text?
There are a lot of parentheses without a space before throughout all the document.
Reviewer 3 Report
Based on postoperative pathology, we conducted a detailed study on the diversity and differences of lung microbiota in patients with pulmonary nodules.
There are differences in pulmonary microbiota between patients with lung cancer and benign pulmonary nodules.
The diversity of microbiota in patients with lung cancer is lower than that in benign pulmonary nodules.
There were significant differences in microbiota among lung cancer patients with different pathological types and smoking status.
The changes in the abundance of lung microbiota are related to the development of lung adenocarcinoma with different histopathological features.
The differential bacterial species found in this study, especially Streptococcus_mitis, may be used as potential biomarkers for lung cancer, and also closely related to the development of invasive adenocarcinoma.
Our study has made a complete picture and analysis of the lung flora in patients with pulmonary nodules, and our findings provide new targets for the diagnosis and treatment of lung cancer, pending our further basic research.
The quality and content of all the figures are excellent

Round 2
Reviewer 1 Report
the authors have responded satisfactorily to my observations.
